# Ultra close-range digital photogrammetry in skeletal anthropology: A systematic review

**Paolo Lussu***[◦], **Elisabetta Marini** *[◦]

Department of Life and Environmental Sciences, University of Cagliari, Cagliari, Italy

◦ These authors contributed equally to this work.
* emarini@unica.it (EM); paolo.lussu@gmail.com (PL)

## Abstract

### Background

Ultra close-range digital photogrammetry (UCR-DP) is emerging as a robust technique for 3D model generation and represents a convenient and low-cost solution for rapid data acquisition in virtual anthropology.

### Objectives

This systematic review aims to analyse applications, technical implementation, and performance of UCR-DP in skeletal anthropology.

### Methods

The PRISMA guidelines were applied to the study. The bibliographic search was performed on March 1st, 2019 using Scopus and MEDLINE databases to retrieve peer-reviewed studies accessible in English full-text. The authors worked independently to select the articles meeting inclusion criteria, upon discussion. Studies underwent to quantitative and qualitative syntheses.

### Results

Twenty-six studies were selected. The majority appeared in 2016 or after and were focused on methodological aspects; the applications mainly dealt with the documentation of skeletal findings and the identification or comparison of anatomical features and trauma. Most authors used commercial software packages, and an offline approach. Research is still quite heterogeneous concerning methods, terminology and quality of results, and proper validation is still lacking.

### Conclusions

UCR-DP has great potential in skeletal anthropology, with many significant advantages: versatility in terms of application range and technical implementation, scalability, and photorealistic restitution. Validation of the technique, and the application of the cloud-based approach, with its reduced requirements relating to hardware, labour, time, and cost, could further facilitate the sharing of large collections for research and communication purposes.

**Data Availability Statement:** All relevant data are within the paper.

**Funding:** This work was supported by the Open Access Publishing Fund of the University of

Cagliari, with the funding of the Regione Autonoma della Sardegna – L.R. n. 7/2007 to EM.

**Competing interests:** The authors have declared that no competing interests exist.

## Introduction

Virtual anthropology is best characterised as an interdisciplinary field of research, mainly developed over the past two decades to study anatomical data representations in 3D. Its major benefits relate to the application of non-invasive procedures to obtain virtual specimens for powerful descriptive, comparative and functional morphological studies [1–3]. Such potential enables a wide use within the scientific community and the general public, including preservation, materialisation and sharing.

A number of different techniques and procedures have been developed to achieve accurate and reliable 3D models of anthropological specimens, such as computed tomography (CT), magnetic resonance imaging, laser scanning, structured light scanning, and digital photogrammetry, in association with various software [2,4]. However, CT and laser scanning require expensive equipment, intricate workflows, and trained operators, and therefore are resource-intensive [3,5,6]. Structured light scanning could be implemented through low-cost hardware and software, but its accuracy is not sufficient for skeletal anthropology applications [7,8].

Digital photogrammetry enables 3D reconstructions from digital photographs of the object [9,10] (see panels 1 and 2 on principles and historical development). Different from satellite and aerial photogrammetry, respectively based on remote sensing images and aerial photographs, it is applied in terrestrial contexts, which include a wide range of potential subjects. When the camera-object distance is under 300 m, it is generally referred to as close-range digital photogrammetry (CR-DP) [10]. The denomination of ultra close-range digital photogrammetry (UCR-DP) is suggested here for cases within a working distance of 10 m, suitable for anthropological subjects and their in situ documentation.

UCR-DP is emerging as a robust technique for 3D model generation and represents a convenient and low-cost solution for rapid data acquisition [11,12]. In fact, some methodological studies reported the best practices [6,13,14], and UCR-DP has already been widely applied in fields closely related to the anthropological research, such as archaeology and cultural heritage–for the surveying, interpretation and virtual reconstruction of excavation sites, caves, buildings, monuments [9,12,15,16], documenting statues, bas-relief and mosaics [17,18], building 3D repositories of museum collections [19]–and palaeontology, particularly for site interpretation and track site documentation [13,20,21], other than for digitising bones from mounted skeletons [22]. A number of applications have also been developed in anthropology for documenting rock art [23–27], artefacts [28,29], cut and percussion marks due to human activity [30–34], and hominin footprints [35].

This review aims to analyse technical implementation, applications and performance of ultra close-range digital photogrammetry in virtual anthropology, focusing on skeletal anthropology. To our knowledge there are no systematic reviews on this specific subject, although a few studies have summarised selected literature including photogrammetry among other 3D techniques in relation to the application of advanced techniques in virtual anthropology [3], forensic anthropology and taphonomy [5,6], and in situ documentation of skeletal remains [9].

### Panel 1: Principles and methods of digital photogrammetry

Photogrammetry encompasses mathematical methods in order to derive information concerning the size, shape, and location of an object from one or more photographs. Following the mathematical model of the central projection imaging, the coordinates of the object surface are estimated by identifying the homologous features in two or more images taken from different perspectives [10].

Because photogrammetry uses light as the information carrier, it is included within non-contact, optical measurement methods, in the class of triangulation techniques, which provide

information only related to the external surface of an object [10]. Unlike terrestrial laser scanning or structured light scanning, photogrammetry is a passive technique that relies on the ambient light reflected by the specimen rather than actively obtaining range data [9]. When applied to produce computer representations, photogrammetry falls into the field of digital image-based modeling (IBM) techniques, allowing the creation of 3D models using data from two-dimensional images [9]. Ultra close-range digital photogrammetry (UCR-DP) represents a variant of CR-DP, indicated to reconstruct objects within a working distance of 10 m. CR-DP and UCR-DP can be further categorised depending on where the software for their implementation resides. Offline photogrammetry relies on locally installed software and on the hardware provided by the user, while cloud-based software environments host the processing logic and data storage capabilities into remote servers operated by a third-party cloud services provider.

As well as CR-DP, UCR-DP workflow encompasses three main phases: shooting, mesh processing (including sparse and dense point cloud generation; mesh and texture construction [9]), and mesh post-processing.

*Shooting* relates to the specimen photographic documentation. The shooting protocol should be carefully planned [1,13] in accordance with both photographic principles [36] and specimen characteristics. In fact, improper shooting affects the quality of the outcome, causing noise or topological artefact because the geometric information acquired from the specimen is insufficient or that from the background is excessive [37]. Therefore precautions should be taken so as to capture the maximum amount of the specimen geometry by enhancing image resolution and depth of field, and by ensuring adequate image coverage, framing, and shooting environment.

To enhance image resolution, digital single-lens reflex (DSLR) cameras with a full-frame sensor and high definition prime lenses are best; ISO sensitivity should be set to the minimum, and precautions for image stabilisation should be taken, such as tripod mount, remote shutter release, or self-timer [36].

For achieving an adequate depth of field the lens diaphragm should be closed by increasing the f-numbers until the whole specimen is in focus, as any further increase would only reduce image resolution. Then, the camera should be set to aperture priority mode (A or Av), where the desired f-number is given, and the shutter speed is chosen accordingly, based on the lighting conditions [36].

Moreover, it is good practice to match the specimen's apparent maximum dimensions with the frame size, while taking the necessary precautions to avoid any cropping. In fact, the closer the camera is positioned to the object, the more detail can be obtained [8,13]. Conversely, if the specimen dimensions are too small compared to the frame size, its geometric features could be ignored by UCR-DP algorithms, and the use of macro-lenses and short shooting distance is necessary. As UCR-DP algorithms extract geometrical features from their perspective change, the specimen should be oriented so as to maximise the detectable change.

With respect to the shooting environment, a well-conceived plain and out of focus background isolates the subject, reducing the need for time-demanding masking interventions in the following phases. Moreover, regular and diffused illumination abates the areas of shade over the specimen's surface, thus preventing the loss of geometric information and non-corresponding image features between the perspective views. Lastly, an appropriate setting of the white balance is necessary to render colours and texture appearance faithfully.

*Mesh processing* starts with feature correspondence and structure-from-motion algorithms matching the homologue points between the images, calculating the camera pose and calibration without prior information, and generating a sparse point cloud that describes the main geometric features of the specimen surface. From this data, multi-view stereo algorithms build

a dense point cloud representing the external surface of the object in detail [9]. Then, the geometry of the specimen is built connecting the dense points and generating a polygon mesh of millions of faces. The appearance and colours of the object are obtained as a texture from the source images and superimposed to the mesh to originate a photorealistic 3D model [9,11].

UCR-DP meshes require to be carefully scaled in order to embed absolute dimensions to them. This step will affect the accuracy of all subsequent measures. One or more linear distances should be measured on the actual specimen, and then the measured value should be referred to the same distance on the 3D model. Alternatively, scaling could also be achieved through calibration markers or millimetric scale bars being recorded in the shots, as some off-line commercial software packages allow scaling of the model through reference distances located on the input photographs.

*Mesh post-processing* concerns itself with the improvement of the quality of the 3D models to make them usable for research and communication. In particular, post-processing is useful to simplify the geometric representation of the mesh in order to make it easy to be visualised, studied, or materialised.

Regrettably, there is a lack of standardisation in terminology, as the term *photogrammetry* itself could also refer to linear measurements obtained from photographs [38–40] or, inexactly, to active structured light techniques [41–43]. More confusion is arising as the expressions *structure from motion* and *dense image matching*–which identify peculiar algorithms applied in the geometry reconstruction–and *computer vision* are becoming increasingly used to refer to the technique itself [8,14,44,45].

## Panel 2: Historical development of digital photogrammetry

Photogrammetry applications started as early as photography itself when, in the 1840s and 1850s, Aimé Laussedat succeeded in measuring buildings from perspective views [46,47]. The technique was then introduced into cartography, topography, architecture and archaeology, mainly for landscape surveys [12], and passed through four main phases of technical development–*plane-table*, *analogue*, *analytical and digital* [10]. The latter two saw the first applications in skeletal anthropology.

*Analytical photogrammetry* took place after the 1950s, with the availability of the first generation of programmable digital computers. A *stereo comparator* was used to measure the parallax between corresponding points in two photographs or X-rays taken from different perspectives. Point coordinates were then inputted into an electronic computer returning the data to produce elevation maps where contour lines connected points with the same elevation [10,48].

In 1980 the first application to skeletal anthropology was illustrated when human skulls were photographed from various perspectives and data were integrated into a coordinate system by means of a computer [49,50].

Implementations and procedures changed dramatically after the introduction of *Digital photogrammetry*, involving fast, digital image processing. In 1993 an early example of UCR-DP application to skeletal anthropology involved the capture of the 3D surface of bone metaphyses and joints from image pairs acquired with 256x256 pixel resolution through an 8-bit digitiser. Dedicated software was developed to process the data and display the three-dimensional surfaces [51]. Just a few years later the availability of significantly more powerful graphic workstations led to a major increase in the complexity of processable data. In 1998 and 1999 UCR-DP was used to reconstruct parts of the glacier mummy known as "Ötzi" the Iceman [52].

The development of sophisticated structure-from-motion and multi-view stereo algorithms represented a major breakthrough which allowed the extraction of 3D data and texture from unordered images of unknown calibration and poses [9]. More recently, cloud-based digital photogrammetry has allowed 3D models to be obtained via the Internet.

# Methods

## Review protocol

**Search strategy and eligibility criteria.** The study was carried out following the *Preferred Reporting Items for Systematic reviews and Meta-Analyses* (PRISMA) guidelines [53]. Inclusion criteria considered the source type, incorporating peer-reviewed studies related to UCR-DP and skeletal anthropology, accessible in English full-text. The bibliographic search was performed on March 1$^{st}$, 2019, using of Scopus and MEDLINE online databases.

The search query was applied to the source title, abstract, and keywords, and included combinations of at least one of the terms identifying the field of application (i.e.: *anthropology*, *anthropometry*, *paleoanthropology*, and *palaeoanthropology*) with at least one of the terms correctly identifying the technique (i.e.: *photogrammetry*, *photogrammetric*, *stereophotogrammetry*, and *stereophotogrammetric*). Furthermore, in order to reduce the risk of bias due to the use of imprecise terminology, the following terms were also included in the query: *structure from motion*, *SFM*, *dense image matching*, *DIM*, *shape from stereo*, *SFS*, and *videogrammetry*. The resulting search query was:

*(\*ANTHROPO\*) AND ((\*PHOTOGRAMM\*) OR ("STRUCTURE FROM MOTION") OR ("SFM") OR ("DENSE IMAGE MATCHING") OR ("DIM") OR ("SHAPE FROM STEREO") OR ("SFS") OR ("VIDEOGRAMMETRY"))*

An iterative process was followed to identify progressively, in greater and greater detail, the scientific papers relevant to the present review (Fig 1). After duplicates removal, the first screening considered the source type and title of the publications retrieved by the search query and classified them as relevant or not. To confirm the fit with eligibility criteria, the second screening considered the abstract of any relevant study and the third involved full-text reading.

Finally, to widen the analysis, both a backward citation analysis (i.e.: the screening of selected articles references) and forward citation analysis (i.e.: the screening of latest studies quoting the selected articles, using Google scholar) were performed. The newly identified articles underwent, recursively, the same citation analysis.

At any step, both the authors worked independently to select the articles meeting inclusion criteria, upon discussion.

**Data collection and paper analysis.** Relevant articles were analysed to extract details on:

- bibliographic features: authors, journal, journal area and category (according to SCImago), year and country of publication, research group countries, scientific journal ranking (SJR, for studies which appear in SCImago);

- study type: methodological (including reviews) or application studies;

- study characteristics: aim, the field of application, sample, technical implementation (photogrammetric technique; hardware and software; shooting protocol; mesh processing, post-processing, and analysis), quality of results (3D model characteristics and accuracy; data validation procedures and results; statistics), availability or online sharing of 3D models.

Relevant studies underwent a qualitative synthesis considering the following key questions.

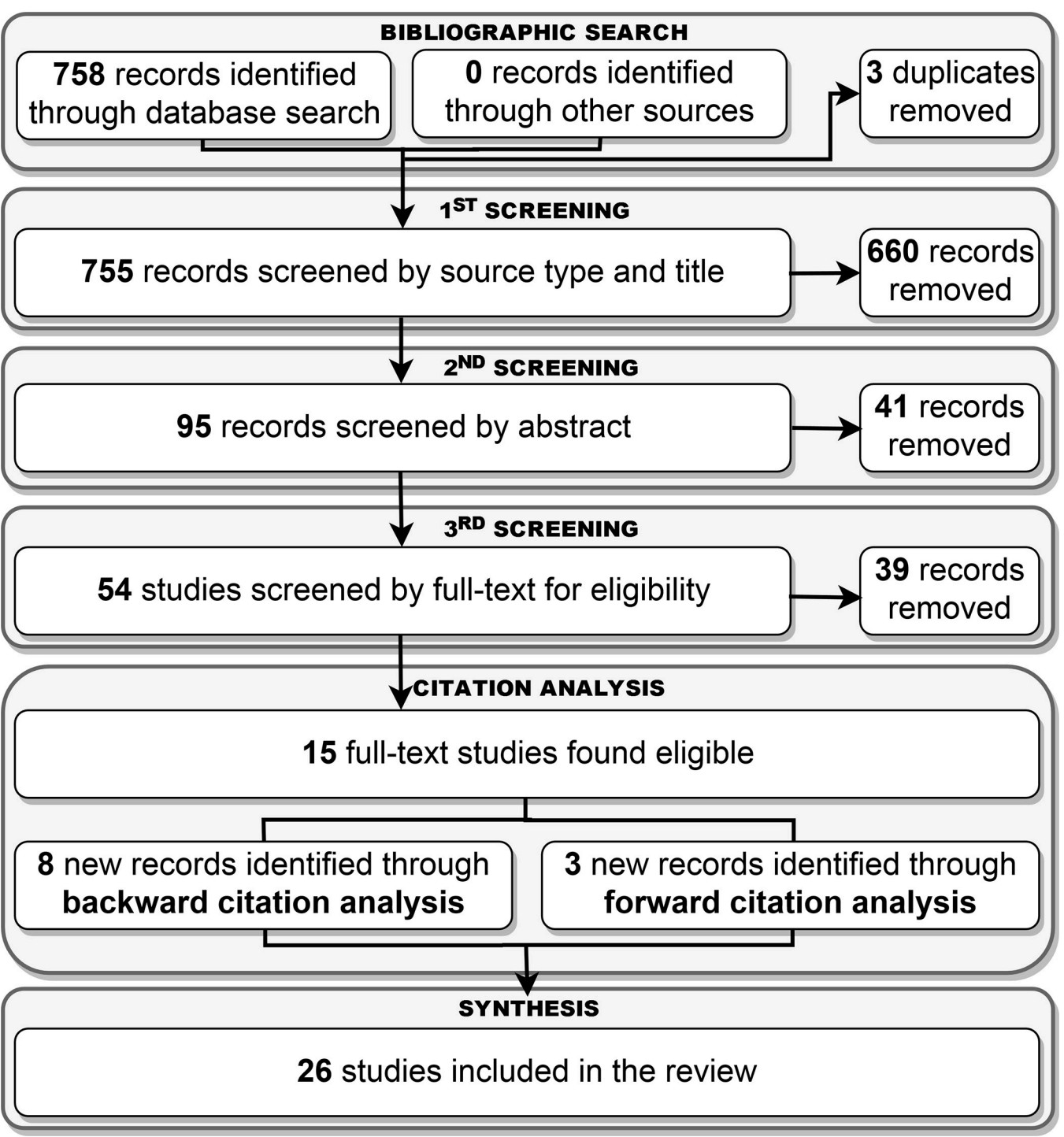

**Fig 1. Flow chart for the study identification.**

1. What are the applications of UCR-DP in skeletal anthropology?

2. What are the most used technical implementations?

3. What is the accuracy and reliability of the technique, and how was it assessed?

## Results

### Search yield and bibliographic characteristics

The bibliographic search retrieved 758 documents from the Scopus online database (Fig 1). MedLine did not contribute any additional documents. Three duplicates were removed before continuing the analysis. Following the examination of source type, title and keywords, 95 articles were found to be possibly relevant. On the basis of the abstract reading, 54 articles underwent full-text reading; of them, 15 were found to fulfil the eligibility criteria. Nine additional records were identified through *backward-* and 3 through *forward-citation analysis*. Therefore, a total of 26 articles were included in the review.

The articles included in the review were published from 2010 to 2019 (Table 1). The majority of them appeared in 2016 or after. Most of the research was conducted in Europe (especially in the United Kingdom and in Italy) and in the Americas (especially in the United States).

According to SJR, the journals where the studies have been published are of medium or high impact. The areas of the journals were quite heterogeneous, with the majority of them falling into the SCImago areas of 'Social sciences' (mostly 'Anthropology' and 'Archaeology' categories) and 'Medicine' (mostly 'Pathology and Forensic Medicine' and 'Anatomy' categories) (Fig 2).

The majority of studies (77%) were primarily of methodological concern, including book chapters and narrative reviews. Despite an increasing trend in the most recent years (Fig 3), studies applying UCR-DP in skeletal anthropology remain scarce, as the most widely used techniques for 3D reconstruction continue to be CT and laser scanning [2,3,5,54].

Methodological research included:

- comparative studies against various alternative methods [8,11,14,55–57];

- comparison of different approaches for morphological analysis [12,58];

- introduction of new procedures with specific purposes: integrating archaeological and osteological data [44,59–62]; identifying carnivore agents on skeletal remains [63]; body identification and forensic facial reconstruction [56,64]; diaphyseal cross-sectional measurement [65]; automatising data acquisition via a microcontrolled turntable [57];

- realisation of a dataset for diagnostic purposes [8].

The narrative reviews and book chapters were related to the general application of different 3D imaging and reconstruction methods in different fields (physical anthropology [3]; forensic anthropology and taphonomy [5,6]; in situ human remains recording [9,60]).

### Qualitative analysis

**1) What are the applications of UCR-DP in skeletal anthropology?**

The applicative studies mainly dealt with the documentation of skeletal findings [55,66,67], including their taphonomy [44,62]; the identification or comparison of anatomical features and trauma [8,58,63,68]; the use of three-dimensional printing techniques for communication and educational purposes [69]. Their focus was related to fossil remains within the human lineage until the early development of anatomically modern humans, hence falling into the field of Palaeoanthropology [12,57,63,69], prehistoric and historic samples [55,59–62,66,67,70], or with contemporary skeletal remains [5,6,68,8,11,14,37,56,58,64,65].

**2) What are the most used technical implementations?**

**Table 1. Summary of the reviewed articles.**

| JOURNAL AREAS (AND CATEGORY) [a] | FIELD OF APPLICATION (STUDY TYPE)– MAIN AIM | SAMPLE | AUTHORS' COUNTRIES | REFERENCE |
|---|---|---|---|---|
| **Medicine** (Pathology and Forensic Medicine) | **Human biology (Methodological)**– Introducing a methodology for diaphyseal cross-sectional measurement | **1 humerus, 1 femur, and 1 tibia** *Modern skeletal collection (Athens, Greece)* | Greece | **Bertsatos & Chovalopoulou (2019)** [65] |
| **Medicine** (Anatomy) **Social Sciences** (Anthropology) | **Human biology (Methodological)**–Assessing best practices for producing 3D digital cranial models | – | United Kingdom | **Morgan et al.(2019)** [14] |
| **Engineering** (Electrical and Electronic Engineering; Mechanical Engineering); **Physics and Astronomy** (Instrumentation) | **Palaeoanthropology (Methodological)**– Improving 3D data acquisition via a micro-controlled turntable | **Saccopastore 1 Neanderthal skull** *Museo di Antropologia della Sapienza (Roma, Italy)* | Italy | **Buzi et al.(2018)** [57] |
| **Medicine** (Pathology and Forensic Medicine) **Biochemistry, Genetics and Molecular Biology** (Genetics) | **Forensic anthropology (Methodological)**– Assessing different 3D printers and software settings in reproducing cranial traumas | **1 human and 1 pig crania** | Canada | **Edwards & Rogers (2018)** [37] |
| **Arts and Humanities** (Conservation) | **Human biology (Methodological)**–Reviewing 3D imaging techniques in virtual anthropology | – | Italy | **Profico et al. (2018)** [3] |
| – | **Taphonomy (Methodological)**–Reviewing techniques in taphonomy | **Human and non-human skeletal remains** | United Kingdom / South Africa | **Randolph-Quinney et al. (2018)** [b] [6] |
| **Computer Science** (Information Systems) **Social Sciences** (Geography, Planning and Development) | **Prehistoric and historic anthropology (Application)** Documenting a finding and its excavation process | **Skeletal remains from 3 burials** *Amiternum medieval site (L'Aquila, Italy)* | Italy | **Trizio et al. (2018)** [67] |
| **Agricultural and Biological Sciences** (Ecology, Evolution, Behavior and Systematics); **Earth and Planetary Sciences** (Earth-Surface Processes, Oceanography, Palaeontology) | **Palaeoanthropology (Methodological | Application)**–Introducing a methodology for identifying the agent of carnivore tooth pits | **OH8 and OH35 hominids** *Olduvai Gorge (Tanzania)* | Spain / South Africa | **Aramendi et al. (2017)** [63] |
| **Arts and Humanities** (Archaeology, History) **Social Sciences** (Archaeology) | **Prehistoric and historic anthropology (Application)** Documenting a finding | **L2A skeleton** *Cussac cave (France)* | France | **Guyomarc'h et al. (2017)** [55] |
| **Arts and Humanities** (History) **Social Sciences** (Archaeology) | **Palaeoanthropology (Methodological)**– Comparing landmark- and high-density point clouds-based approaches to describe morphology | **9 mandible casts** from the Homo lineage | United Kingdom | **Hassett & Lewis-Bale (2017)** [12] |
| **Medicine** (Pathology and Forensic Medicine) | **Forensic anthropology (Methodological)**– Introducing a technique for human body identification | **13 skulls** | Italy | **Santoro et al. (2017)** [64] |
| – | **Human biology (Methodological)**–Reviewing the application of digital-based modeling to the recording of in situ human remains | – | United Kingdom | **Ulguim (2017)** [b] [9] |
| **Medicine** (Pathology and Forensic Medicine) | **Forensic anthropology | Taphonomy (Methodological)** Proposing a new approach in mass grave documentation and study | **6 teaching skeletons** | United Kingdom | **Baier & Rando (2016)** [44] |
| **Computer Science** (Applications; miscellaneous) | **Human biology (Methodological | Application)**–Producing a 3D dataset of the lumbar spine vertebras, and validating the method | **86 human lumbar vertebrae (10 for the validation)** *Trotter Anatomy Museum (Dunedin, New Zealand)* | New Zealand | **Bennani et al. (2016)** [8] |
| – | **Forensic anthropology (Methodological)**– Reviewing the applications of digital imaging in forensic anthropology | – | United States | **Garvin & Stock (2016)** [c] [5] |
| **Nursing** (Issues, Ethics and Legal Aspects) **Medicine** (Pathology and Forensic Medicine) | **Human biology (Methodological | Application)**–Testing the landmark and mesh-to-mesh approaches in assessing sex and ancestry | **80 human adult crania** | Czech Republic | **Jurda & Urbanová (2016b)** [58] |
| **Social Sciences** (Anthropology) | **Prehistoric and historic anthropology (Methodological)** Addressing issues and limitations in photogrammetry and laser scanning | **Skeletal elements from 3 Upper Palaeolithic individuals** *(Dolní Věstonice, Czec Republic)* | Czech Republic | **Jurda & Urbanová (2016a)** [70] |

*(Continued)*

**Table 1.** (Continued)

| JOURNAL AREAS (AND CATEGORY) [a] | FIELD OF APPLICATION (STUDY TYPE)–MAIN AIM | SAMPLE | AUTHORS' COUNTRIES | REFERENCE |
|---|---|---|---|---|
| **Medicine** (Anatomy) **Social Sciences** (Anthropology) | **Prehistoric and historic anthropology (Methodological \| Application)** Integrating excavation and post-processing data from archaeological and osteological contexts | **2 skeletons from the Migration Period** (AD 400–550, Sandby ring fort, Öland island, Sweden) | Sweden | **Wilhelmson & Dell'Unto (2015)** [62] |
| **Arts and Humanities** (Arts and Humanities (miscellaneous)); **Social Sciences** (Archaeology) | **Prehistoric and historic anthropology (Methodological)** Summarising the development of reflexive methods at Çatalhöyük | **Various skeletal remains from unspecified burials** (Çatalhöyük, Konya, Turkey) | Sweden United States United Kingdom | **Berggren et al. 2015)** [60] |
| – | **Prehistoric and historic anthropology (Methodological)** Integrating tools and methods to make the excavation process virtually reversible, thus helping human burials interpretation | **Various skeletal remains from unspecified burials** (Çatalhöyük, Konya, Turkey) | United States Sweden | **Forte et al. (2015)** [c] [61] |
| – | **Forensic anthropology (Methodological)** Demonstrating a protocol for forensic facial reconstruction | **1 cranium** | Italy Brazil | **Morales et al. (2014)** [c] [56] |
| – | **Prehistoric and historic anthropology (Methodological)** Integrating tools and methods to make the excavation process virtually reversible, thus helping human burials interpretation | **Various skeletal remains from unspecified burials** (Çatalhöyük, Konya, Turkey) | United States | **Forte (2014)** [d] [59] |
| **Medicine** (Anatomy) **Social Sciences** (Anthropology) | **Human biology (Methodological)**–Assessing UCR-DP in capturing and quantifying human skull morphology | **4 modern crania of Mongolian origin** Musée de l'Homme (Paris, France) | United States France | **Katz & Friess (2014)** [11] |
| – | **Palaeoanthropology (Application)**–Capturing National Museums of Kenya and Turkana Basin Institute's collections in digital format to be accessed on-line and interacted with | **Various specimens of the Homo lineage** National Museums of Kenya and Turkana Basin (Kenya) | Kenya United States | **Leakey & Dzambazova (2013)** [b] [69] |
| **Computer Science** (Computer Graphics and Computer-Aided Design; Human-Computer Interaction) **Engineering** (miscellaneous) | **Prehistoric and historic anthropology (Application)**–Documenting a finding | **51 skulls and associated skeletons from the Anglo-Saxon period** (AD 910–1030, Weymouth, United Kingdom) | United Kingdom | **Ducke et al. (2011)** [66] |
| **Medicine** (Anatomy) **Social Sciences** (Anthropology) | **Compared anatomy (Application)**–Studying trapeziometacarpal joint curvature among five extant Primates genera, including Homo | **58 trapezia and 58 first metacarpals**, plus other specimens from other present and past species | United States | **Marzke et al. (2010)** [68] |

a According to Scimago.

b Peer-reviewed book chapter.

c Study published in a peer-reviewed journal not indexed in Scopus or Scimago.

d Study published in a peer-reviewed journal indexed in Scopus or Scimago after 2017.

**Shooting.** Sixteen studies [6,8,58,63,64,67,68,70,11,12,14,37,44,55–57]–over 23 producing 3D models–specified the shooting protocol, while other studies provided fragmentary information (Table 2).

**Mesh processing.** All but one [69] study used dedicated 3D model production computer hardware. Powerful graphic workstations [8,11,14,37,44,56,70], up to a set of 64 cores and 512 GB RAM [8], were used.

Studies employed offline software packages, with a single exception in which a cloud-based environment was adopted, ReCap Photo (Autodesk Inc., USA) [69] (Table 3). Most authors used a commercial solution–mainly PhotoScan (Agisoft, Russia) [6,8,60–62,67,70,11,12,14,37,44,55,57,59] or PhotoModeler (EOS Systems, Canada) [64,68]–while

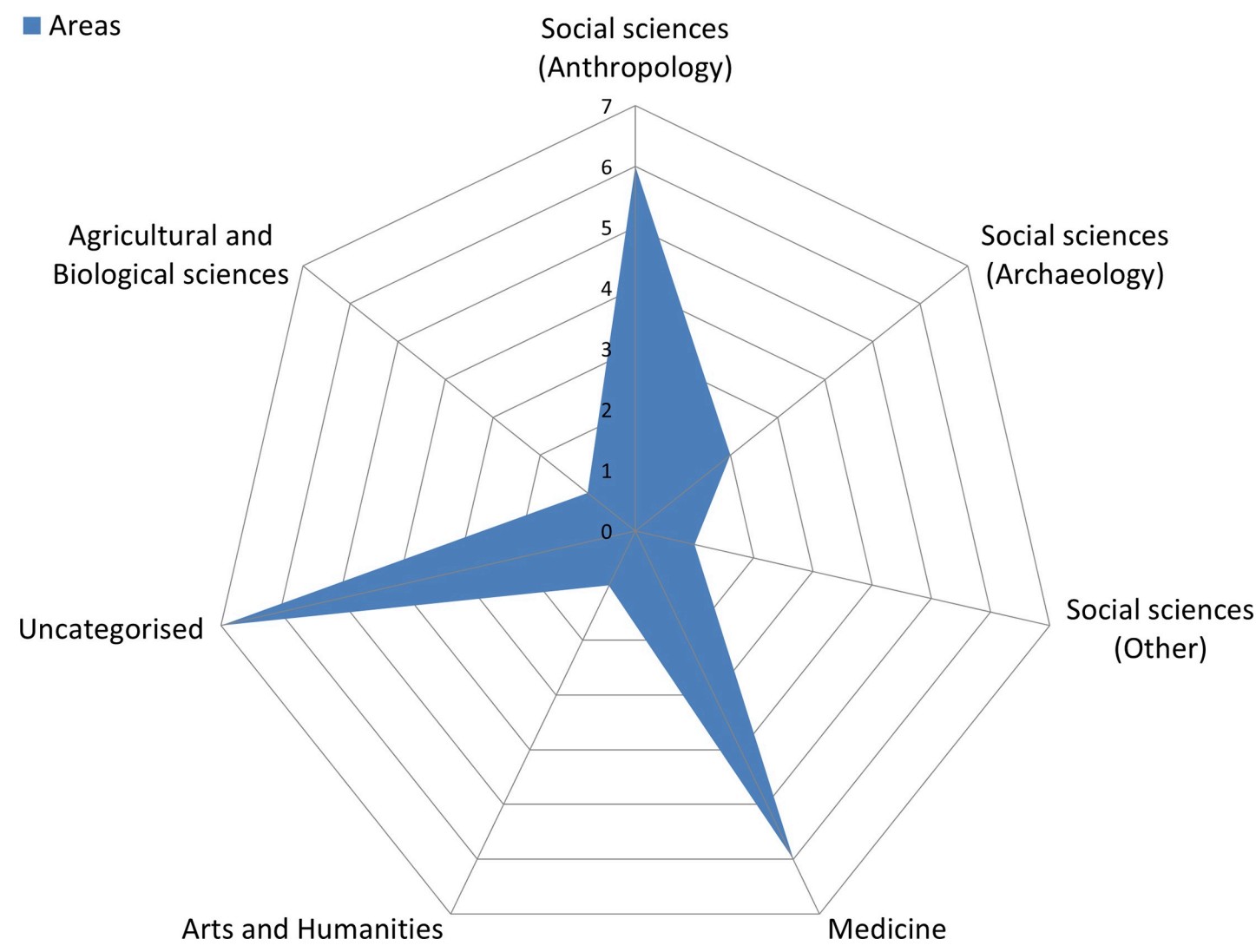

**Fig 2. Distribution of the publication journals within SCImago disciplinary areas.**

others used a combination of commercial and open-source or free for academic use software packages, such as PhotoScan and PMVS (Patch-Based Multi-View Stereo Software [71]) [55], or applied open-source or free for academic use solutions, such as GRAPHOS (inteGRAted PHOtogrammetric Suite [72]) [12,63]; Bundler and MeshLab (Visual Computing Lab, ISTI-CNR, Italy [73,74]) [66], or PPT-GUI (Python Photogrammetry Toolbox with Graphic User Interface [75]) and MeshLab [56], or did not give any information [65].

Within the studies adopting discrete shooting sessions (Table 2), the alignment and merging of the partial meshes for 3D reconstruction was realised using MeshLab [56,70], PhotoScan [55,57], or CloudCompare [63]. Different algorithms were applied, such as the least-squares optimization [63] and the iterative closest point (ICP) [8,12,70] for the alignment, and the Poisson remeshing algorithm [70] for the fusion.

File formats for input and output data were declared in a few studies. For input photographs it was mainly JPEG [14,56,57,69]; in one case this was accompanied by the occasional use of lossless camera proprietary RAW [69]. For the final 3D processing outcome PLY [56,66], OBJ [64,65], and 3D PDF [59] file formats were employed.

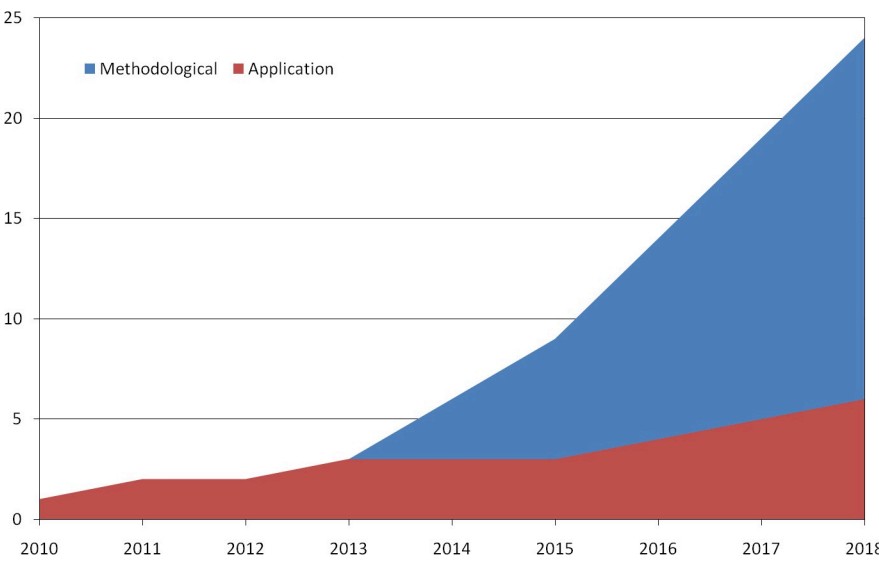

**Fig 3. Classification of reviewed studies according to their aim. data up to December 31st, 2018.**

Scaling details were reported by a minority of the reviewed studies. The procedure was based on reference scales included in the frame [55,58,63,68,70], or on linear measurements [11,12,56,57] taken on the actual specimen and then applied to the 3D model. The measures were taken a single time [56,57], or were repeated [11,12]; in all cases along one axis only. The software packages more frequently used for mesh scaling were MeshLab [37,56,70], PhotoScan [11,14], and Geomagic Studio (3D Systems Inc., United States) [11].

As for the 3D model production time for a single specimen, the shooting session required up to 60 minutes [56], while the manual correction of the photographic masks took on average about 1 min per image [11]. The mesh processing required a minimum of 70–80 minutes [63], between 110 and 300 minutes [8,11], and up to 540 minutes [56].

**Mesh post-processing.** Post-processing issues were discussed by some authors from a general perspective, highlighting the problems that may arise, and the best practices to prevent them [6,11,56,58,70]. A wide variety of software was applied (Table 3). Open-source software packages such as MeshLab [8,56,58,59,61,66,70] and CloudCompare [12,56,58,69] were the most used in many contexts. Some studies used commercial software for specific tasks: Photo-Scan [67]; Geomagic Studio [11], and Geomagic Wrap (3D Systems Inc., United States) [55,70].

**Mesh analysis.** Depending on the aim, 3D model analysis was carried out using several software packages (Table 3). Open-source solutions such as CloudCompare [12,56,58,69] were widely applied for mesh orientation and comparison [12,37,56,58,69], and for data analysis [37,44,61]. MeshLab was used similarly for visualisation and manipulation [62,64,66], and, along with CloudCompare, for data analysis [44,61]. For landmarking and measurement Avizo Software (Thermo Scientific, United States) [63], ArcScene (Esri, United states), and NewFaceComp (which was developed ad hoc for the study) [64] were used. TIVMI (PACEA laboratory, Université Bordeaux 1, France), specifically developed for skeletal anthropology applications, was frequently applied for the same tasks.

For comparative analyses, free software Morphologika [12] was used to calculate geometric morphometric distances while open-source software packages Meshlab [37], FIDENTIS Analyst [58] and CloudCompare [12,37,44,56], along with commercial Geomagic Studio [11], were all used for mesh-to-mesh and mesh-to-point cloud comparisons.

**Table 2. Shooting protocol summary.**

| CAMERAS | | |
|---|---|---|
| *Type* | **APS-C DSRL** [6,8,67,69,70,11,12,44,55,56,58,63,64]—*of which, with integrated GPS module*: [67] | **Other** (ultra-compact [37]; tablet PC camera [59]; C-mount microscope camera [68]) |
| *Number* | **One** [6,8,63,64,67–70,11,12,37,44,55,56,58,59] | |
| **LENSES** | | |
| *Type* | **Zoom** (standard [12,37,55–57]) | **Prime** (standard [8]; macro [58,63,70]) |
| *Focal length* | **Fixed** (<50mm [8,55]; 50mm [12]; >50mm [56,58,70]) | **Variable** [6,11,37] |
| **SHOT SETTINGS** | | |
| *Exposure* | **Given ISO** (100 [57], 400 [8]) | **Variable ISO** [6,11,37]; [14] [a] |
| | **Given diaphragm aperture** (f/8 [8], f/10 [57], f/22 [70], f/27 [58], f/32 [56]) | **Variable/automatic diaphragm aperture** [6,11,12,37,44,55,63,64]; [14] [a] |
| | **Given time** 1/30s [8], 1/2s [57], 3s [58] | **Variable time** [14] [a] |
| **SHOOTING ENVIRONMENT** | | |
| *Specimen installation* | **Turntable** (manually operated [6,8,11,12,14,37]; microcontrolled [57]) | **Fixed support** (styrofoam ring [56]) or **in situ documentation** [a] [44,55,66,67] |
| | Distance of 45 cm [57], 50 cm [14] cm from the camera lens | |
| *Background* | **Plain and uniform** [6,8,14,57] –*of which, with pre-shooting masking* [14] | |
| *Illumination* | **Constant** [6,8,14,56–58,63,64]—*of which, with a lightbox* [57] | **Variable** (flash [14] [a]) |
| *Stabilisation* | **Tripod mount** [8,56–58,63,64] | **Other** (Remote shutter release [56,58]; self-timer [58]) |
| **SHOOTING PROCEDURE** | | |
| Shooting sessions | All studies reported sequential shooting sessions, i.e. the production of consecutive shoots | |
| | **Continuous** [b] [6,11,67–70,12,14,37,44,58,59,64,66] | **Discrete** [c] (manual merging [8,56,63,70]; automatic merging [55,57]) |
| *Shoot n°* | < **85** [11,12,37,64,68,70]          **85–120** [58,63,65] | > **120** [8,14,57,67,69] |
| *Image overlap* | **Horizontal** (7˚ [57], 10˚ [8,37], 12˚ [56], 15˚ [63], no more than 30˚ [66], 30˚ [68]) | **Vertical** (15˚ [8], 20˚ [63], 30˚ [57], 35˚ [56]) |
| *Perspective change* | **Specimen rotation** [6,8,11,12,14,37,57] | **Camera movement** (rotation around the specimen [56,58,63]; other [68]; in situ [d] [44,55,59,66,67]) |
| *Duration* | **10 minutes** [11], **30 minutes** (70), **60 minutes** [56] | |

[a] Used automatic camera mode, although recommending aperture priority mode.

[b] I.e. a continuous series of shoots to obtain a single complete mesh directly.

[c] I.e. discrete series of shoots to obtain several partial meshes to be merged.

[d] No study specified the pattern followed for in situ documentation.

### 3) What is the accuracy and reliability of the technique, and how was it assessed?

Although no study was openly aimed to validate UCR-DP in skeletal anthropology, some authors provided quantitative data on its outcome assessment, with respect to other techniques such as osteometry [8,14,55,59]; 3D coordinate digitiser [8]; laser scanning [11,56], or microtomography [57] (Table 4). The aim of such studies included the assessment of UCR-DP performance in: describing the human skull [11] [14] and vertebrae [8]; in situ measurement [55,59], and facial reconstruction [56]. The suitability of specific procedures, such as the use of a micro-controlled turntable [57], was also considered. Furthermore, a study validating 3D printing procedures using different hardware and software settings included data produced via UCR-DP [37], among other techniques. However, in this latter case, as all of the data sources were evaluated altogether, no specific result for UCR-DP was recognisable.

Study models varied. When declared, the sample included 1 [55–57] to 10 [8] specimens, while measured variables ranged from 5 [8] to 50 [14]. Only in three cases did authors declare that measurements had been taken under osteological criteria [11,14,55].

**Table 3. Software used for mesh processing, post-processing, and analysis.**

| SOFTWARE (Producer, author or reference) | LICENCE | APPLICATION AND REFERENCES | | |
|---|---|---|---|---|
| **PhotoModeler** (EOS Systems, Canada) | Commercial | **Processing** | | **Offline 3D reconstruction** [64,68] |
| **GRAPHOS** ([72]) | Open-source | | | **Offline 3D reconstruction** [12,63] |
| **Patch-Based Multi-View Stereo** ([71]) | – | | | **Offline 3D reconstruction** [55,66] |
| **Bundler** (Snavely, Noah) | Open-source | | | **Offline 3D reconstruction** (sparse point cloud generation) [66] |
| **PPT-GUI** ([75]) | Open-source | | | **Offline 3D reconstruction** (dense point cloud generation) [56] |
| **PhotoScan** (AgiSoft, Russia) | Commercial | | **Post-processing** | **Offline 3D reconstruction** [6,8,60–62,67,70,11,12,14,37,44,55,57,59] |
| | | | | **Decimation, remeshing and hole filling** [67]; **Scaling** [11,14] |
| **ReCap Photo** (Autodesk Inc., United States) | Free for academic use | | **Analysis** | **Cloud-based 3D reconstruction** [69] |
| **MeshLab** (Visual Computing Lab, ISTI-CNR, Italy [73,74]) | Open-source | | | **Offline 3D reconstruction** (in support of) [56]; **Optimisation** [59] |
| | | | | **Noise reduction, cleaning and hole-filling** [8,56,58,66,70] |
| | | | | **Partial meshes alignment and fusion** [56,70] |
| | | | | **Scaling** [37,56,70]; **Visualisation, manipulation** [62,64,66] |
| | | | | **Data analysis** [44,61] |
| **CloudCompare** (Developed by Girardeau-Montaut et al. since 2003) | Open-source | | | **Offline 3D reconstruction** (in support of) [12,63] |
| | | | | **Mesh alignment and comparison** [12,37,44,56,58,69] |
| | | | | **Data analysis** [44,61] |
| **Geomagic Studio** (3D Systems Inc., United States) | Commercial | | | **Noise reduction, and hole-filling** [11]; **Scaling** [11] |
| | | | | **Segmentation** [11] |
| **Geomagic Wrap** (3D Systems, Inc., United States) | Commercial | | | **Segmentation** [55,70] |
| **ArcScene** (Esri, United states) | Commercial | | | **Visualisation and manipulation, in situ** [62] |
| | | | | **Landmarking and measurement, in situ** [62] |
| | | | | **3D georeferencing and geo-spatial analysis** [62] |
| **Avizo Software** (Thermo Scientific, United States) | Commercial | | | **Landmarking and measurement** [5,59,63] |
| **TIVMI** (PACEA laboratory, Université Bordeaux 1, France) | – | | | **Landmarking and measurement** [11,55,70] |
| **FIDENTIS Analyst** ([86]) | Open-source | | | **Mesh comparison** [58] |
| **Morphologika** (Developed by O'Higgins and Jones, 2006) | – | | | **Mesh comparison** [12] |

In fact, most studies provided data on accuracy, where UCR-DP showed similar results compared to osteometry [14], laser scanning [11] [56] and CT scanning [57]. The bias in relation to the reference technique was generally under 2 mm and 2%. However, performance was slightly worse when the technique was applied to relatively small specimens such as vertebrae [8], and in situ [55,59]. In this latter case high-end laser scanners have been found to outperform UCR-DP for surveys [59].

As for the reliability, only two studies included repeated measurements in their model [11] [14], while the coefficient of variation, intraclass correlation coefficient, standard error of measurement, combined standard uncertainty of measurement [76], along with inter-observer data, have not been investigated yet. Moreover, while a few studies provided the standard deviation for the whole sample [11,55], reliability comparing repeated measurements of the same

**Table 4. Summary of literature on UCR-DP accuracy and reliability.**

| STUDY MODEL | | | RESULTS | | REFERENCE |
|---|---|---|---|---|---|
| **SAMPLE AND MEASUREMENT** | **REFERENCE TECHNIQUES** | **STATISTICAL TECHNIQUES** | **ACCURACY** | **RELIABILITY** | |
| 3 crania– 50 measures 3 repetitions– 1 rater | Osteometry | Bland–Altman | Bias < 2 mm (2%) [a] | – | Morgan et al. (2019) [14] |
| | | | Bias range: 0.11 to 1.93 mm [bc] | | |
| | | | Bias range %: 0.84 to 2.82 [bc] | | |
| | | | LOA (best): -0.96 to 1.74 mm [c] | | |
| | | | LOA (worst): -1.14 to 1.87 mm [c] | | |
| 1 cranium– 16 landmarks No repetition– 1 rater | CT-scanning | Geometric morphometrics | Bias: 1.6 mm | – | Buzi et al. (2018) [57] |
| | | | Bias range: 0.43 to 3.08 mm [d] | | |
| 1 cranium– 30 measures No repetition– 1 rater | Osteometry (in situ) | – | Bias: 2.4% (0.01–7.9%) | – | Guyomarc'h et al. (2017) [55] |
| 10 vertebrae– 5 measures No repetition– 1 rater | Osteometry Arm-scanning | Bland–Altman | Bias: 5.2% [e], 4.7% [f] | – | Bennani et al. (2016) [8] |
| | | | Bias < 3.5 mm [f g] | | |
| | | | LoA: -4.4–5.4 mm [e], | | |
| | | | LoA: -4.8–5.0 mm [f] | | |
| – | Osteometry (in situ) | – | Bias: ~5 mm | – | Forte (2014) [59] |
| 4 crania– 16 landmarks 2 surface areas 4 repetitions– 1 rater | Laser scanning | Geometric morphometrics ANOVA | "Bias < 2 mm [h] | 112 mm² [ij] 9.6 mm² [ik] | Katz & Friess (2014) [11] |
| | | | Bias + 1.2% [ij] | | |
| 1 cranium No repetition– 1 rater | Laser scanning | – | Bias: ± 1 mm | – | Morales et al. (2014) [56] |

[a] In most cases.

[b] Data for 3D models created using 150 or more photographs.

[c] Data for 3D models created on high- or ultra-high alignment and dense point cloud software settings.

[d] Range of the absolute landmark displacement between UCR-DP and CT-scanning.

[e] Data referred to osteometry.

[f] Data referred to arm scanning.

[g] In 95% of measures.

[h] Data relative to linear measurements.

[i] Data relative to surface area measurements.

[j] Replication error in measuring parietal bone area.

[k] Replication error in measuring nasal bone area.

variable on the same specimen was rarely studied, and only one study provided some data on UCR-DP reliability in measuring bone surface areas [11], showing a slightly better performance of UCR-DP compared to laser scanning.

## Discussion

Within the rich and articulated scenario of three-dimensional reconstructions in skeletal anthropology, UCR-DP represents the least used method. However, its utility and suitability have stimulated a growing interest, particularly in the last three years.

Probably due to the novelty of the application of UCR-DP in the field, the research is quite heterogeneous concerning methods and quality of 3D results. Moreover, it is mainly of methodological concern, aimed at describing or comparing procedures or new possible

applications, while application study numbers remain inadequate and do not show a relevant increase over the time [8,55,66–69]. However, some pieces of research, mainly dealing with methodological aspects, include the application of the procedure to real cases [58,62,63].

1) What are the applications of UCR-DP in skeletal anthropology?

The field in which UCR-DP has been more frequently employed is in situ documentation of skeletal remains [9], aiming to describe the specimen [55], the taphonomic processes [62], and the phases of the excavation process [44,59–62], or for communication purposes [66]. Such privileged application is attributable to the higher versatility and reduced time requirements of UCR-DP, compared to other surface scanning or range techniques. Photogrammetry is more practical in sites presenting access limitations for physical or normative reasons [55], and enables 3D reconstructions even when such application has not been planned [66].

Skeletal remains have been contextualised within their environment, be it a burial [62,66,67], an entire settlement [59–61], or a cave with access restrictions [55]. In this context, UCR-DP has sometimes been used to reconstruct skeletal remains [55,61], while other technologies, such as laser scanning, were used for a wider scale survey of the site [59,60]. However, UCR-DP itself is suitable for both purposes, as exemplified by studies reconstructing remains altogether with their burial using site photographic documentation taken ad hoc [59,60,62,67] or making the most of a pre-existing photographic archive [66]. UCR-DP data has also been contextualised with other three-dimensional or archaeological sources within the framework of geographic information systems (GIS) [59–62,67], which is particularly useful in documenting complex sites, such as Catalhöyuük [59–61].

Other studies focused on forensic and taphonomic applications, where UCR-DP's ability to capture the surface texture and colour is invaluable for completing a description of the specimen, e.g. in accurately documenting fracture patterns or modifications such as sun bleaching and soil staining [5]. Forensic and taphonomic analyses have also been directed to diagnose sex [58], reconstruct facial morphology [56], help in human body identification [64], document trauma [37], point post-mortem bone fracture patterns out [62,67], identify aspects of past human life and environment, e.g. the carnivore agent who caused death or looted the corpse afterwards [63]. Indeed, UCR-DP 3D models are suitable in a criminal investigation. However, the lack of standardised and validated protocols still negatively affects their probative value as court evidence in legal proceedings [5,6].

Another field of application includes comparative studies, aimed at determining the individual's ancestry group [58] or the similarities among fossil hominins and extant catarrhine genera [68]. Indeed, this is promising, although an almost unexplored application of photogrammetry in skeletal anthropology. In fact, while comparative studies have already been performed on 3D models reconstructed from CT and laser scanning data, sometimes comparing them with UCR-DP-derived data obtained from other studies [54], the extensive application of UCR-DP would enable researchers to study considerably wider samples.

Because of their ease of production, and photorealistic textures, UCR-DP 3D models are particularly suitable for dissemination, the creation of large databases, visualisation, and materialisation. A set of guidelines has been proposed for three-dimensional digital data publication [77], and several online repositories are available for indexing 3D data, often allowing also its storage, and therefore the sharing of 3D models without the need for the creation and maintenance of a dedicated website (Table 5). However, existing digital skeletal collections are mainly based on data sources other to UCR-DP, such as CT or MRI.

Among the aforementioned repositories MorphoSource is the world's most popular one for research purposes, [78], while Sketchfab is considered a de-facto standard for publishing 3D content on the web [79]. MorphoSource uses creative commons licences, but it lacks immediacy of use and interactivity, not allowing the content to be visualised and manipulated online,

**Table 5. Online repositories for skeletal 3D models sharing.**

| NAME Website | MAIN AIM | FIELD | ACCESS RIGHTS | LICENSING | 3D SCANNING TECHNOLOGIES | UCR-DP MODELS | CONTR | VIEW | DOWN | SHARE | EMBED | MUSE |
|---|---|---|---|---|---|---|---|---|---|---|---|---|
| **3D Cad Browser** *www.3dcadbrowser.com* | Commercial | Various | Open, or for a fee | Chosen by contributor | Various | Some | ✓ | ✓ | ✓ | ✗ | ✗ | ✗ |
| **3D Virtual Museum** [b] *http://www.3d-virtualmuseum.it* | Non-commercial | Cultural Heritage | Open | Chosen by contributor | UCR-DP, LS, SLS | Some | ✓ | ✓ | ✗ | ✓ | ✓ | ✗ |
| **Africanfossils** [c] *https://africanfossils.org* | Non-commercial | Palaeontology | Open | CC BY-NC-SA | UCR-DP, SLS | Most | ✗ | ✓ | ✓ | ✓ | ✓ | ✓ |
| **ARIADNE Visual Media Service** [d] *http://visual.ariadne-infrastructure.eu* | Non-commercial | Cultural Heritage | Open | CC BY-NC | Various | Most | ✓ | ✓ | ✓ | ✓ | ✓ | ✗ |
| **Digimorph** *http://www.digimorph.org* | Non-commercial | Biology | Open for personal use | Chosen by contributor | CT | #x2013; | ✓ | ✓ | ✓ | ✗ | ✗ | ✗ |
| **Digital Archive of Fossil Hominoids** *https://www.virtual-anthropology.com/virtual-anthropology/share/digital-archive-of-fossil-hominoids* | Commercial | Palaeoanthropology | For a fee | #x2013; | CT | #x2013; | ✗ | ✗ | ✗ | ✗ | ✗ | ✗ |
| **Digital Morphology Museum, KUPRI** *http://dmm.pri.kyoto-u.ac.jp/dmm* | Non-commercial | Primates | Open for educational and research purposes | #x2013; | CT, MRI | #x2013; | ✗ | ✓ | ✓ | ✗ | ✗ | ✗ |
| **Dryad** [d] *https://datadryad.org* | Non-commercial | Various | Open | Chosen by contributor | Various | Some | ✓ | ✓ | ✓ | ✗ | ✗ | ✗ |
| **Figshare** [d] *https://figshare.com* | Non-commercial | Various | Open | Chosen by contributor | Various | Some | ✓ | ✗ | ✓ | ✓ | ✓ | ✗ |
| **GB3D Type Fossils** [e] *http://www.3d-fossils.ac.uk* | Non-commercial | Palaeontology | Open | CC BY-NC-SA | Various | Some | ✓ | ✓ | ✓ | ✗ | ✗ | ✗ |
| **MorphoMuseuM** [d] *https://morphomuseum.com* | Non-commercial | Vertebrates | Open for educational and research purposes | CC BY-NC | Various | #x2013; | ✓ | ✓ | ✓ | ✗ | ✗ | ✗ |
| **MorphoSource** *https://www.morphosource.org* | Non-commercial | Biology | Open | CC BY-NC [f] | Various | Some | ✓ | ✓ | ✓ | ✗ | ✗ | ✗ |
| **Nespos** [g] *https://www.nespos.org* | Non-commercial | Primates | Open | Chosen by contributor | CT | #x2013; | ✓ | ✗ | ✗ | ✗ | ✓ | ✗ |
| **Phenome10** *https://phenome10k.org* | Non-commercial | Biology | Open | CC BY-NC | Various | #x2013; | ✓ | ✓ | ✓ | ✗ | ✗ | ✗ |
| **Sketchfab for cultural heritage** [d] *https://sketchfab.com/museums* | Hybrid | Various | For a fee | Chosen by contributor | Various | Some | ✓ | ✓ | ✓ | ✓ | ✓ | ✓ |
| **Smithsonian Natural History Museum** [h] *http://humanorigins.si.edu/evidence/3d-collection* | Non-commercial | Biology | Open | #x2013; | Various | Some | ✗ | ✓ | ✓ | ✗ | ✗ | ✗ |

*(Continued)*

Table 5. (Continued)

| REPOSITORY | | | | DATA | | | FUNCTIONS [a] | | | | | |
|---|---|---|---|---|---|---|---|---|---|---|---|---|
| NAME Website | MAIN AIM | FIELD | ACCESS RIGHTS | LICENSING | 3D SCANNING TECHNOLOGIES | UCR-DP MODELS | CONTR | VIEW | DOWN | SHARE | EMBED | MUSE |
| SpineWeb [g] http://spineweb.digitalimaginggroup.ca | Non-commercial | Human spine | Open for research purposes [i] | Chosen by contributor | X-ray, CT, MRI | #x2013; | ✓ | ✗ | ✗ | ✗ | ✗ | ✗ |
| Turbosquid http://www.turbosquid.com | Commercial | Various | Open, for a fee | Royalty-free | Various | #x2013; | ✓ | ✗ | ✓ | ✗ | ✗ | ✗ |
| Virtual Fossils [b] http://www.virtualfossils.com | Non-commercial | Palaeontology | Open | CC-BY | CT, mostly | #x2013; | ✓ | ✓ | ✗ | ✓ | ✓ | ✗ |
| Zenodo [b] https://zenodo.org | Non-commercial | Various | Open | Chosen by contributor | Various | #x2013; | ✓ | ✗ | ✓ | ✓ | ✗ | ✗ |

[a] Detail of the functions allowed for three-dimensional data: CONTR, contribution; VIEW, online viewing; DOWN, downloading; SHARE, social media sharing; EMBED, embedding in external websites; MUSE, virtual museum online.

[b] Database links to Sketchfab resources.

[c] Example of a virtual museum, linking to Autodesk online resources.

[d] Suitable for research purposes.

[e] Restricted to reference specimens for species description.

[f] Recommended licensing.

[g] Database links to external repositories.

[h] Download allowed upon request.

[i] After contributor's agreement.

or embedded into external websites, although registered users are allowed to download data. On the other hand, Sketchfab is a commercial solution and lacks the flexibility to meet the diversified needs of the cultural heritage field; furthermore, it uses lossy compression, and consequently most of the 3D models found there are drastically simplified in their geometry [80].

Platforms specifically suitable in skeletal anthropology are the ARIADNE Visual Media Service (VMS) [80], aimed at supporting cooperative work in archaeology by the sharing of large visual data, and MorphoMuseuM [81], meant to improve the knowledge of vertebrate fine anatomy. They are both based on 3DHOP (3D Heritage Online Presenter, Visual Computing Lab, ISTI–CNR, Italy) [80,82], an open-source software package for the online presentation of data in the Cultural Heritage field.

It is noteworthy that existing digital skeletal collections include a limited number of specimens. Indeed, as highlighted in a recent *Nature* analysis, palaeontologists are reluctant to share their data [78]. Accordingly, among the reviewed studies, only some authors provided their repository upon request [8], or shared their reconstructions temporarily, in situ, during short-term exhibitions open to the public–such as for an interactive mass burial reconstruction [66]–or to support the ongoing research at Catalhöyük, aiming to experiment with an immersive environment for research and educative purposes [59–61]. A noticeable exception to the aforementioned picture is represented by the African Fossils Project (http://www.africanfossils.org) [69], where a friendly environment reproduces a virtual lab in which 3D embedded UCR-DP models of specimens from the National Museums of Kenya and the Turkana Basin Institute can be explored, downloaded, or shared by social media functions. Unfortunately, the 3D models are made available in low resolution only.

Significant issues limiting the open access to 3D models are related to the rarity of the specimens, the intensive resources associated with their scanning and post-production [60], and the constraints deriving from intellectual property rights [77,78]. Researchers traditionally do not share data about their ongoing or future work because of their fear of receiving insufficient acknowledgement by scientists who use it. Museums are also concerned with sharing data from the collections in their care because of economic reasons and copyright policies. The sharing of anthropological remains is also limited by ethical and political reasons, such as in the case of remains originating from indigenous people, who generally do not approve their publication [78]. Whatever the reasons, such limitations surrounding the free sharing of three-dimensional data reduce the opportunities for science communication, and hence the potential for scientific knowledge evolution. However, some of the mentioned limitations could be overcome by using an easier technique for 3D reconstruction, such as UCR-DP, and specific copyright conditions, such as a creative commons licence (https://creativecommons.org).

Indeed, there is a clear tendency towards an increase of palaeontological 3D data sharing: several museums have recently rewritten their policies, and many journals and professional societies are encouraging it [78]. It is noteworthy that the Archaeology Data Service of the United Kingdom has developed guidelines detailing the good practice for preservation and documentation of 3D models in Archaeology (https://guides.archaeologydataservice.ac.uk/g2gp/3d_Toc).

2) What are the most used technical implementations?

**Shooting.** The shooting protocol should be planned carefully (see panel 1) as any mistake in this phase cannot be rectified without repeating the data acquisition [6]. Instead, within the 16 studies reporting the shooting protocol, only a few described it in detail. Most of them limited the information to the type and number of the cameras applied, the number of shooting sessions and shots, and the way of changing the specimen's perspective view. Hence, the appropriateness of the procedures used for enhancing image resolution and depth of field, fit

the specimen image to the frame size, and install a suitable light environment, was frequently difficult to ascertain.

As for the resolution, no author reported the use of a DSLR with a full-frame sensor, a few studies mentioned the adoption of prime lenses [8,58,63,70], ISO sensitivity was set to the minimum in one case only [57], and precautions for image stabilisation were rarely declared [8,56–58,63,64]. Some authors declared the use of zoom lenses [12,37,55–57], even if they usually show higher optical aberrations and lower optical resolution than the prime ones [36].

As for the depth of field, only a few authors set a narrow diaphragm aperture [56,58,70]. Others left the choice to the camera [6,11,12,14,37,44,55,63,64], a sub-optimal option that enhances image resolution at the expense of the achievable depth of field [36]. However, it should be noted that wide diaphragm settings could be used without contraindications for in situ documentation [44,55,62,66,67], where the wider distance between the subject and the lens allows for sufficient depth of field regardless of the diaphragm aperture [36].

As for the framing, among the reviewed studies, only two specified the distance between the camera lens and the specimen [14,57], and three specified the use of macro-lenses [58,63,70].

The shooting environment also implies appropriate specimen installation, background, Illumination, and image stabilisation. To hold a skeletal specimen, a convenient option, frequently used in the reviewed studies, is that of a turntable [6,8,11,12,14,37,57], i.e. a rotating platform facilitating the shooting phase and improving its repeatability. Such platforms should be of uniform colour and texture, matching those of the background, so as not to provide additional geometric information, and could be operated either manually or automatically. Specimen installation could also be facilitated by the use of a styrofoam ring, as done in one study [56]. Despite their relevance, only a minority of the authors declared their concerns with assuring an appropriate light environment [6,8,14,56–58,63,64] and a plain uniform background [6,8,14,57], using, for instance, a lightbox [57] or a white cloth placed underneath the rotating platform [14]. One of the reviewed studies found that 3D models outcomes were not affected by the use of an inconstant light source such as a camera-mounted flash [14]. Besides, no research declared the use of white balancing procedures nor specified light source colour temperature.

Another fundamental issue concerns image stabilisation, i.e. the prevention of any specimen or camera shake, so as to avoid motion blurs that cause a loss of detail and incorrect shot alignment [37,83]. Among the reviewed studies, clearer images were achieved by mounting the camera on a tripod [8,56–58,63,64], and using a remote shutter release [56,58], or a self-timer [58].

All studies reported sequential shooting sessions, instead of simultaneous ones, i.e. the production of shots from different perspectives using multiple cameras, that would have sped up the image acquisition process (Table 2). Some of the reviewed studies [6,11,67–70,12,14,37,44,58,59,64,66] performed sequential shooting sessions with a continuous approach, whereas others used the discrete procedure. In the latter case a subsequent manual [6,11,67–70,12,14,37,44,58,59,64,66], or automatic [55,57] alignment and merging of the partial meshes was needed, thus implying a longer procedure.

Besides requiring more time, the discrete approach has been found associated with a decay in the quality of the outcomes [70]. Indeed, the fusion of partial meshes introduces additional sources of error due to the higher degree of subjectivity in the alignment, thus resulting in a less accurate 3D model [8,37,70]. Furthermore, using the Poisson remeshing algorithm to merge the partial meshes produced extensively smoothed surfaces with localised defects [70]. For the above-mentioned reasons, the discrete approach should be avoided whenever possible [6].

Another relevant factor affecting the quality of the final product is the number of photos used to generate the model. This aspect was highly variable within the reviewed literature, as the shots taken for a specimen ranged from 3 to 320. According to a recent technical note, the optimal number for the skull is around 150 [14]. In fact, beyond this number there is a significant increase in model creation time and no detectable improvement. On the other hand, while using as few as 50 images is sufficient to reconstruct a complete cranium, the corresponding 3D models showed poor quality [14]. A recent study (Lussu et al., submitted) found that 100 shots are sufficient for the purpose where the protocol is conceived so as to achieve additional perspective views from the regions showing higher geometrical complexity. In fact, orienting the specimen's axis of symmetry with the lowest order, perpendicularly to the horizontal camera plane, enables it to capture the maximum geometric information when changing the perspective view, and thus reduces the required number of poses. However, a large image overlapping should be ensured. Indeed, this is achievable by using an angular difference between consecutive shots generally not greater than 15˚ along the horizontal plane [8,37,56,57,63] and 35˚ along the vertical one [8,56,57,63]. It should be noted that in situ documentation generally requires a higher number of shots, due to the extent of the recorded area [67].

It is remarkable that no author controlled the data acquisition via software. The open-source software package digiCamControl (http://digicamcontrol.com) enables multiple camera management for simultaneous or sequential shooting sessions, and automatic image indexing and storage, thus reducing the possibility of errors and the time demand and cost of the process (Lussu et al., submitted).

## Mesh processing

Almost all the studies used an offline approach based on commercial software, while the use of open-source [12,55,56,63,66] or cloud-based free for academic use [69] software were the exceptions. This has several disadvantages that go beyond the cost of software acquisition and updates. Offline computing requires powerful computational hardware with high-end processors and a great amount of working memory. Furthermore, the use of offline software packages is time-demanding, as they engage the processing capabilities of the local hardware for several hours, and follow many steps to generate a mesh and its texture. Even where open-source solutions were employed, this was usually done through a combination of offline software packages, each specific for any of the reconstruction steps [12,55,56,63,66], sometimes requiring the use of additional general-purpose software, such as MeshLab [56] or CloudCompare [12,63], for completing mesh reconstruction. A few free and open-source software packages are available for offline processing–such as Regard3D (http://www.regard3d.org) and MeshRecon (http://zhuoliang.me/meshrecon.html)–even if they have not been validated in skeletal anthropology yet, and hence their suitability is unknown.

Regarding the input images, it has been shown that UCR-DP algorithms are very good at dealing with differences in resolution, exposure or lighting conditions [14]. In fact, algorithms return the best performance in reproducing fine details when using lossless file formats, including TIFF or camera proprietary RAW [13,14,66]. However, photographs should never be cropped, as this would change the relative scale of the image features retrieved from different shots [66]. Raw input images have their drawbacks in requiring more computational resources, and therefore, when the maximum accuracy is not the priority, such as for dissemination, high-quality JPEG input is justifiable [13], and suitable to create accurate photogrammetric models [56].

Among the reviewed studies, the scaling of 3D models has been based both on calibration scales and markers [55,58,63,68,70], or on linear distances on the specimen [11,12,56,57]. In

the latter case, as suggested by Hassett & Lewis-Bale [12], a distance between arbitrary reference stitches is preferable to standard osteological measures whose landmarks can be more difficult to be accurately and precisely localised [12]. Once the measurements had been taken, scaling was usually achieved via the 3D reconstruction software, although other software, such as MeshLab [70], was occasionally used. Regrettably, mesh scaling on the basis of an arbitrary distance is cumbersome with MeshLab, while other software packages, such as ReCap Photo, have the ability to identify the landmarks more easily and precisely (Lussu et al., submitted).

To increase the accuracy of the scaled meshes only two studies [11,12] took repeated measurements, while no study applied a separate scaling factor for each of the three orthogonal axes. A differential scaling along the axes is achievable with MeshLab. However, a recent study has shown that it produces only minor variations (Lussu et al., submitted).

The time required for the offline mesh processing of a single specimen (from 70 to 540 minutes, among the reviewed studies) depends on the complexity of the geometry to be rebuilt, the local hardware specifications, the image number, and the protocol type. For instance, discrete shooting sessions take much longer [70].

As opposed to the offline approach, the cloud-based one only require users to upload the photographs, without prior masking, and to download the three-dimensional outcome at the end of the processing, enabling effective 3D reconstruction in about 20 minutes including the shooting session, scaling and post-processing (Lussu et al., submitted).

## Mesh post-processing

In order to improve the quality of the 3D models, remove artefacts, and simplify mesh geometry, the open-source software MeshLab was the most used in the reviewed literature for its flexibility and wide capability of application to cleaning, trimming, noise reduction, hole-filling, decimation, and remeshing [8,56,58,59,61,66,70]. Commercial software was rarely used for the purpose [11,67].

Despite the recognised benefits of 3D post-processing, an excessive level of intervention should be avoided as it would introduce errors in the final 3D model [66], such as filling authentic holes, or smoothing and down-sampling complex mesh data. However, as highlighted by some authors [3,37], specific applications such as 3D printing, require a so-called watertight mesh, i.e. a mesh whose surface is continuous, and therefore even actual holes, e.g. the foramen magnum, may need filling.

## File formats

The suitability of a file format for 3D models storage depends on their intended use. Where the main application is research, the choice of PLY or OBJ file formats, supported by the most used software packages for geometric data analysis, is advisable. In fact, other formats, such as 3D PDF, are not readable by most software for 3D models analysis. On the other hand, widespread file formats not supporting the embedding of a texture, such as STL, should be avoided when dealing with UCR-DP data.

Where the main application is dissemination, it should be taken into account if it is to allow the download and reuse of the 3D models, or only their online display and use. In fact, to use the locally downloaded 3D models effectively, the installation of specific software, such as MeshLab [62,64,66], is needed. The 3D PDF file format could represent an effective solution [59], as it can be read by Adobe Reader DC, a globally widespread software package. However, a commercial software licence is needed to generate 3D PDF files, (e.g. Adobe Acrobat Pro, Adobe Photoshop, 3Dsystems Geomagic Studio, 3Dsystems Geomagic Design X).

Where only online visualisation and manipulation is planned, the NXZ file format is more suitable, having been specifically designed for the efficient web-based fruition of very large 3D reconstructions, and supporting their lossless compression, embedding, and streaming, i.e. their visualisation and manipulation while the download is still in progress.

If 3D models are intended both for research and dissemination purposes, again the PLY and OBJ file formats could be the most suitable. For instance, the ARIADNE Visual Media Service (Table 5) supports the upload of both PLY or OBJ files, and they are automatically converted to NXZ for their online display.

## Mesh analysis

Within the reviewed studies, the analyses carried out on the 3D models involved visual assessment [62,64,66], segmentation [11,55,70], landmarking and measurement [5,11,55,59,62,63,70], mesh alignment, comparison and inter-3D model distance calculation [12,37,44,56,58,69], and data analysis [44,61]. As for the software packages for 3D model analysis, although the more frequently used were CloudCompare [12,37,44,56,58,61,69] and MeshLab [44,61,62,64,66], some dedicated software could be more adequate for specific tasks. For instance, ReCap Photo allows users to identify landmarks more easily and precisely, despite lacking the variety of functions of MeshLab (Lussu et al., submitted). Moreover, specific software for landmarking and measurement was used by some authors [5,11,55,59,63,70].

Methodologies for comparing 3D models included the well established geometric morphometrics, [12,55,63] which is a landmark-based approach, and the recent dense cloud and mesh-to-mesh approaches [12,58], in which the entire surfaces of the 3D models are compared to each-other to assess specimens distance. This latter approach allows researchers to study a greater amount of morphological data [12,58]. The two approaches have been compared in a study on hominin mandibular variation [12], returning slightly different group membership estimates.

In summary, the wide range of technical implementations available for implementing UCR-DP, the possibility of automating data flow for handling a variable number of cameras in relation to the specimen and sample characteristics, alongside with the availability of diverse outputs in accordance with the planned data usage, all demonstrate the versatility and scalability of the technique.

### 3) What is the accuracy and reliability of the technique, and how was it assessed?

Most studies [8,11,14,37,55–57,59], with only one exception [70], agree that UCR-DP results are comparable to those produced by osteometry [8,11,14], CT scanning [14,57], laser scanning [11,14,37,56], or structured light scanning [37] (Table 4). The bias was generally below the 2 mm threshold, that is considered an acceptable error in osteometry [84]. Moreover, frequency histograms produced for UCR-DP 3D models measurements were unimodal and normally distributed, meaning that errors were basically random [14]. UCR-DP has been acknowledged to reproduce fully recognisable anatomical traits, although some skeletal areas are more prone to having artefacts [11,14]. Moreover, differently to CT, MRI, and some range techniques, the presence of a photorealistic texture is of invaluable help in locating landmarks and fine structures (Fig 4A1 and 4A2). However, when acquiring a human skull with standard lenses, meshes obtainable via UCR-DP are less dense than those derived from a CT scan (Fig 4B1 and 4B2). Therefore, although the detail appears adequate to describe human crania, it could be insufficient to study smaller structures. More research is needed on the subject. In fact, the sub-optimal performance reported by some authors on relatively small specimens could be due to their size compared to that of the frame, particularly where macro lenses have not been adopted [8], or due to the use of discrete shooting sessions [8,63].

## UCR-DP

## CT-scanning

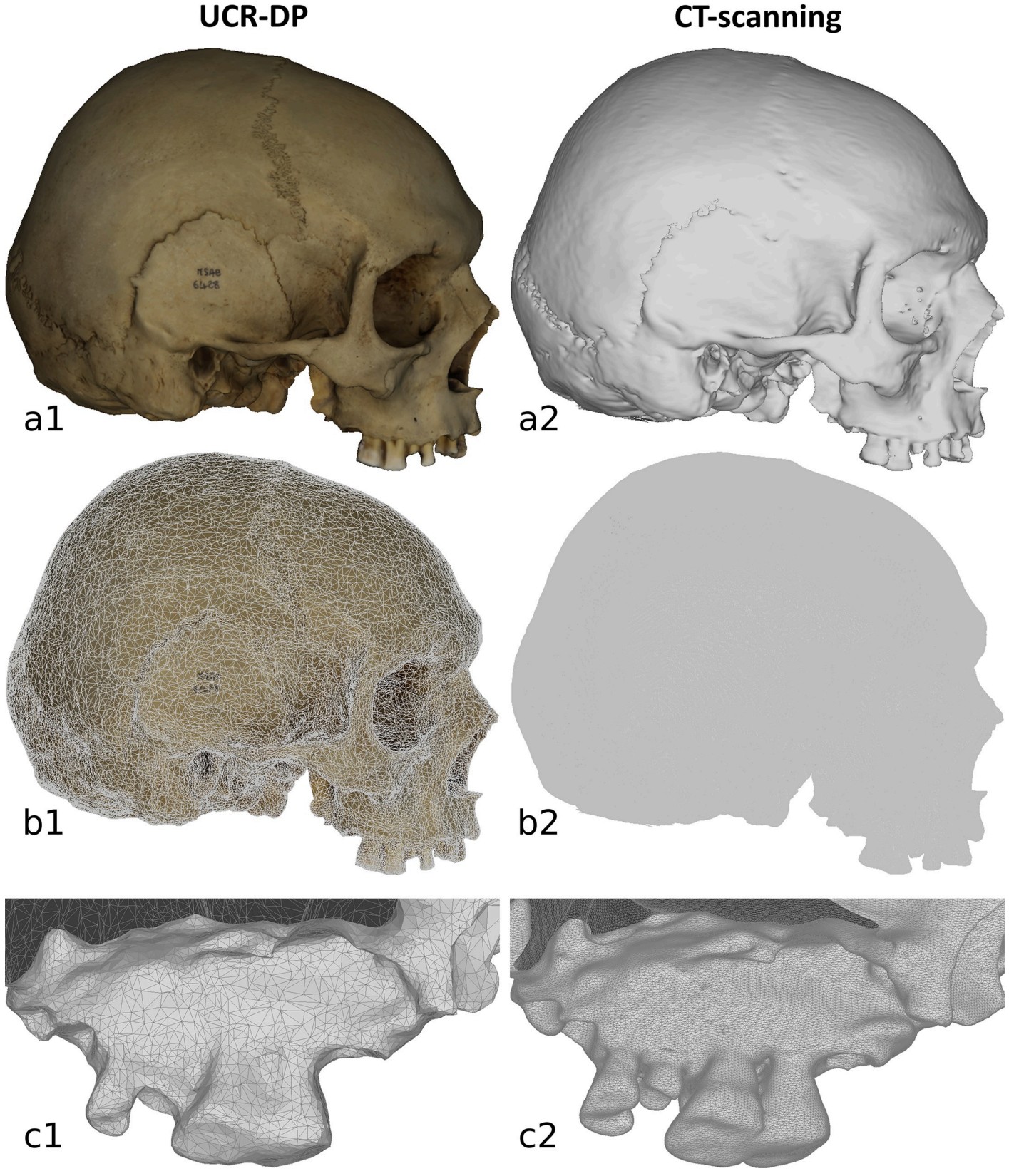

**Fig 4. Comparison between UCR-DP (left) and CT scanning (right) in describing the skull MSAE-6428 (musae—museo sardo di Antropologia ed etnografia, Università degli studi di Cagliari, Italy).** a1-a2. Visual restitution of the whole specimen. b1-b2. Mesh density for the whole specimen. c1-c2. Mesh density detailing the geometry. UCR-DP data collection with two Canon EOS 1200D DSLRs at 100 ISO, using prime 50 mm f/1.8 lenses, 50 cm shooting distance, 5500 K light sources; 3D reconstruction via ReCap Photo cloud-based environment. CT data collection with a Siemens SOMATOM Definition Flash CT-scanner and 0.75 mm slice thickness; segmentation via 3D Slicer 4.8.1 (https://www.slicer.org).

In the case of in-situ documentation, in which the geometric features of the site are recorded jointly with those of the skeletal remains, the scale of the resulting 3D model is so small that accuracy error is likely to rise over the acceptable threshold for osteometric applications [55,59]. In such context, if accuracy is a priority, it is perhaps advisable to use a multimodal method of data collection, using UCR-DP for capturing the skeletal remains at a larger scale and laser scanning for capturing the surrounding environment at a smaller scale [6].

As for the reliability, no study evaluated UCR-DP precision, inter-observer error or combined standard uncertainty of the measurement [76]. Validations assessing the agreement between two techniques with the well standardised Bland-Altman technique [14] are not conclusive if precision is not independently assessed for both techniques by means of repeated measurements of the same variable on the same specimen. In fact, an apparent lack of agreement, or poor agreement, shown by UCR-DP with a reference technique could be the artefactual effect of the measurement imprecision in UCR-DP, in the reference technique, or in both [85]. Our data showed a higher precision for UCR-DP compared to osteometry (Lussu et al., submitted).

In summary, despite the insightful contribution of the reviewed studies, there is still the need for robust validation of UCR-DP, assessing both intra- and inter-observer accuracy and precision against a standardised technique, such as CT-scanning. Furthermore, the validation of software packages other than Photoscan, and that of the cloud-based approach, are completely lacking.

The limitations of this review could be related to an incomplete retrieval of identified research due to the lack of terminology standardisation and improper definition of the technique, still frequent in the literature. However, the inclusion in the search query of a number of terms inappropriately used for referring to the technique, along with the backward and forward citation analyses of selected studies, should have reduced such risk.

Considering the outcome level of the studies, a risk of bias could be linked to the frequently observed poor agreement to basic photographic principles, incomplete description of the protocol, limited sample size, and choice of inadequate statistical techniques.

## Conclusions

UCR-DP offers many significant advantages over other 3D scanning techniques: greater versatility in terms of application range and technical implementation, scalability, and photorealistic restitution. Further benefits include reduced requirements relating to hardware, labour, time, and cost, especially when applying cloud-based and free for academic use solutions. The technique is therefore an attractive option for capturing 3D spatial datasets in skeletal anthropology.

However, despite growing interest, UCR-DP still represents the least used method for three-dimensional reconstruction in skeletal anthropology. Related studies remain mainly of methodological concern, while there are not many actual applications. Most authors used commercial software packages, and an offline approach. The sharing of 3D models was uncommon.

Furthermore, current research is still quite heterogeneous concerning methods, terminology, and quality of results. The protocols for 3D models production, and the relative hardware

are poorly described and not always in agreement with photographic principles and best practices. Indeed, besides some efforts, standardisation of UCR-DP methodologies and protocols, including the cloud-based approach, and validation against reference techniques, such as CT-scanning, is still lacking.

The application of standardised protocols, along with an improved adherence to basic photographic principles during data collection, would level outcome accuracy and reproducibility of future research similar to the best practice studies. Simultaneous analysis of UCR-DP, osteometry, and CT-scanning, performed on the same skeletal sample, under osteometric standards, involving multiple observers, repeated measures, and different types of landmarks, along with the appropriate statistical procedures, would probably be conclusive for the technique reliability. The cloud-based approach could further facilitate the production and open access sharing of large collections for research and communication purposes. Such effectiveness is highly relevant given the amount of undocumented prehistoric and historic skeletal material and sites, especially in low and middle-income countries.

## Supporting information

**S1 Checklist. PRISMA 2009 checklist.**
(DOC)

## Acknowledgments

The authors thank Dominick John Tompkins for proofreading.

## Author Contributions

**Conceptualization:** Paolo Lussu, Elisabetta Marini.

**Data curation:** Paolo Lussu, Elisabetta Marini.

**Formal analysis:** Paolo Lussu, Elisabetta Marini.

**Investigation:** Paolo Lussu, Elisabetta Marini.

**Methodology:** Paolo Lussu, Elisabetta Marini.

**Writing – original draft:** Paolo Lussu, Elisabetta Marini.

**Writing – review & editing:** Paolo Lussu, Elisabetta Marini.

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
