## [Decision Letter · Decision Letter 0]

5 Mar 2020

PONE-D-20-00288

Ultra close-range digital photogrammetry in skeletal anthropology: a systematic review

PLOS ONE

Dear Prof. Marini,

Thank you for submitting your manuscript to PLOS ONE. After careful consideration, we feel that it has merit but does not fully meet PLOS ONE’s publication criteria as it currently stands. Therefore, we invite you to submit a revised version of the manuscript that addresses the points raised during the review process.

We would appreciate receiving your revised manuscript by Apr 19 2020 11:59PM. To enhance the reproducibility of your results, we recommend that if applicable you deposit your laboratory protocols in protocols.io, where a protocol can be assigned its own identifier (DOI) such that it can be cited independently in the future. For instructions see: http://journals.plos.org/plosone/s/submission-guidelines#loc-laboratory-protocols

We look forward to receiving your revised manuscript.

Kind regards,

Lynne A Schepartz

Academic Editor

PLOS ONE

Journal Requirements:

Reviewers' comments:

Reviewer's Responses to Questions

**Comments to the Author**

1. Is the manuscript technically sound, and do the data support the conclusions?

Reviewer #1: Yes

Reviewer #2: Yes

2. Has the statistical analysis been performed appropriately and rigorously? 

Reviewer #1: I Don't Know

Reviewer #2: Yes

3. Have the authors made all data underlying the findings in their manuscript fully available?

Reviewer #1: Yes

Reviewer #2: Yes

4. Is the manuscript presented in an intelligible fashion and written in standard English?

Reviewer #1: Yes

Reviewer #2: Yes

5. Review Comments to the Author

Reviewer #1: I very much enjoyed your paper and find it interesting that a synthetic overview of ultra close range digital photogrammetry has not been previously published. I have a list of simple suggestions for your paper:

Line 141 change to …“lack of standardization in terminology”…

Line 159 awkward...please rephrase

Line 200 perhaps replace “science” with “scientific”

Line 240 change to “the Americas” perhaps?

Line 245 “archaeology”…check the spelling throughout the ms. Be consistent.

Line 522 change “speeded” to “sped”

Typographical error on Figure 2 … Anthropology is misspelled

Reviewer #2: The article was well-researched and presented. As a review article of the "state of the field" regarding UCR-DP, the bibliographic search by the authors, detailed within the manuscript were sufficiently detailed to discover the relevant data for their own analysis and presentation. This article will be helpful for researchers using digital photogrammetry for many applications, not solely skeletal analysis. Herein lies the value of this article in presenting a well-researched bibliography but also in identifying avenues for further testing and refinement of the methodology and analyses as well as new directions for processing data. Despite just a few sentences that could be less-awkwardly worded, and one that was missing a subject for the verb, the manuscript was well-written and intelligible.

6. PLOS authors have the option to publish the peer review history of their article (what does this mean?). If published, this will include your full peer review and any attached files.

Reviewer #1: No

Reviewer #2: No

---

## [Author Response · Author response to Decision Letter 0]

6 Mar 2020

Reviewer #1

We would like to thank you for the comments of appreciation. We have revised the manuscript following all your suggestions. 

Line 141: we have changed to “lack of standardization in terminology”…

Line 159: we have rephrased the sentence 

Line 200: we have replaced “science” with “scientific”

Line 240: we have changed to “the Americas”

Line 245: we have checked the consistency of the term “archaeology” throughout the manuscript

Line 522: we have changed “speeded” to “sped”

Figure 2: we have written Anthropology correctly.

Reviewer #2

We feel honoured for your comments. Thank you for your appreciation. We incorporated all your amendments in the revised version of the manuscript.

---

## [Editor Report · Decision Letter 1]

13 Mar 2020

Ultra close-range digital photogrammetry in skeletal anthropology: a systematic review

PONE-D-20-00288R1

Dear Dr. Marini,

We are pleased to inform you that your manuscript has been judged scientifically suitable for publication and will be formally accepted for publication once it complies with all outstanding technical requirements.

With kind regards,

Lynne A Schepartz

Academic Editor

PLOS ONE